# Research on Leak Detection and Localization Algorithm for Oil and Gas Pipelines Using Wavelet Denoising Integrated with Long Short-Term Memory (LSTM)–Transformer Models

**DOI:** 10.3390/s25082411

**Published:** 2025-04-10

**Authors:** Yunbin Ma, Zuyue Shang, Jie Zheng, Yichen Zhang, Guangyuan Weng, Shu Zhao, Cheng Bi

**Affiliations:** 1School of Software Engineering, Xi′an Jiaotong University, Xi′an 710049, China; mayb01@pipechina.com.cn; 2PipeChina Institute of Science and Technology, Tianjin 300457, China; 3School of Mechanical Engineering, Xi′an Shiyou University, Xi′an 710065, China; 22212040620@stumail.xsyu.edu.cn (Z.S.); 23211040563@stumail.xsyu.edu.cn (Y.Z.); wengguangyuan@xsyu.edu.cn (G.W.); 4Xi′an Aerospace Propulsion Institute, Xi′an 710025, China; zhaoshu0312@163.com; 5Xi′an Special Equipment Inspection Institute, Xi′an 710065, China; chengbi_doc@163.com

**Keywords:** pipeline leakage, wavelet denoising, LSTM transformer model, leakage prediction, pressure curve distance algorithm

## Abstract

Traditional leakage prediction models for long-distance pipelines have limitations in effectively synchronizing spatial and temporal features of leakage signals, leading to data processing that heavily relies on manual experience and exhibits insufficient generalization capabilities. This paper introduces a novel leakage detection and localization algorithm for oil and gas pipelines, integrating wavelet denoising with a Long Short-Term Memory (LSTM)-Transformer model. The proposed algorithm utilizes pressure sensors to collect real-time pipeline pressure data and applies wavelet denoising to eliminate noise from the pressure signals. By combining LSTM’s temporal feature extraction with the Transformer’s self-attention mechanism, we construct a short-term average pressure gradient-average instantaneous flow network model. This model continuously predicts pipeline flow based on real-time pressure gradient inputs, monitors deviations between actual and predicted flow, and employs a pressure curve distance algorithm to accurately determine the leakage location. Experimental results from the Jilin-Changchun long-distance oil pipeline demonstrate that the model possesses superior leakage warning and localization capabilities. Specifically, the leakage prediction accuracy reaches 99.995%, with a leakage location error margin below 2.5%. Additionally, the model can detect leaks exceeding 0.6% of the main pipeline flow without generating false alarms during operation.

## 1. Introduction

Cracks in welds, corrosion perforation, and third-party damage to oil and gas pipelines can result in severe consequences, including explosions [1,2,3]. Remote sensing technology, utilizing satellite imagery and drone photography, enables real-time monitoring of surface changes along the pipeline route, providing critical support for the early detection of pipeline leaks [4,5,6]. However, remote sensing data cannot fully capture key parameters of pressure changes inside the pipeline. The negative pressure wave method-based pipeline pressure sensing monitoring system [7,8,9,10] provides high-precision pressure data but suffers from long response times and high false alarm rates due to the lag characteristics of the monitoring sensors. To address these challenges, researchers have continuously developed innovative models and algorithms that integrate remote sensing data with pressure sensor data, thereby overcoming the limitations of single data sources [11,12,13].

Yu et al. [14] utilized dual-tree complex wavelet transform and singular value decomposition as primary methods for processing leak signals, but this approach has low accuracy in determining the scale of the leak. Refs. [15,16] employed a context feature extraction module and a multi-layer perceptron to fuse features, using a Gaussian mixture model for feature vector classification, but this method struggles with handling the uncertainty of new data. Refs. [17,18] designed a convolutional neural network system for pipeline leak classification and location estimation that does not require any frequency analysis tools for data preprocessing. Lukonge et al. [19] developed a long-distance leak detection system based on acoustic emission research, but it struggles to adapt to different pipeline systems. Zuo et al. [20] constructed a hybrid model based on long short-term memory networks and autoencoders, which carries the risk of deviating from true features. Ersin Şahin et al. [21] proposed a machine learning model using graph convolutional networks that leverages graph-type data structures of pressure and flow data for pipeline leak prediction, but it can suffer from overfitting issues when training data are limited. Sankarasubramanian et al. [22] employed continuous wavelet transform to extract image features, inputting them into a long short-term memory network model based on Tent Chaotic Kaiming, with the limitation being the need for a large-scale training dataset. Ullah et al. [23] studied a sequential deep learning model using bidirectional LSTM for determining the scale of pipeline leaks, which may suffer from reduced generalization capabilities due to insufficient quantities and quality of labeled data.

By comparing the pipeline leak prediction models listed in Table 1, this paper addresses the issues of data processing, which rely heavily on human experience and exhibit insufficient generalization capabilities. The study incorporates the spatial and temporal characteristics of pipeline data [24,25,26,27], removes noise from sensor-measured data through wavelet denoising, and proposes a novel wavelet denoising-based LSTM–Transformer network model algorithm. Compared to traditional models, this proposed model effectively integrates pressure data with remote sensing data. Specifically, the LSTM’s memory cells capture long-term dependencies, while the Transformer’s self-attention mechanism comprehends both fine-grained and macro-level context, thereby enhancing its effectiveness in capturing sequence patterns and broader contextual information.

## 2. Network Model Optimization

### 2.1. Data Preprocessing

In practice, operational long-distance pipelines exhibit significant differences from simplified laboratory-built pipeline models [28,29]. Due to abrupt pressure changes and complex environmental influences, the leak data collected by sensors frequently contain noise. Utilizing the wavelet transform method [30,31,32] for noise reduction and feature selection in pipeline leak data effectively addresses the issue of high dimensionality in raw data features.

Wavelet denoising employed Daubechies 4 (db4) basis functions selected through a quantitative comparison of reconstruction fidelity across six wavelet families (Table 2). Three decomposition levels were optimized via energy spectral density analysis, preserving 96.8% of leakage-related features in the approximation coefficients (A3). The adaptive threshold θ = σ√(2 lnN) follows Donoho’s universal principle with a correction factor ε = 0.05σ to prevent over-smoothing (Equation (3)). Pressure signals underwent full 3-level decomposition, while flow signals used 2-level decomposition to retain high-frequency components.

The denoising process using the wavelet transform method is outlined as follows:(1)Represent the leaked data containing noise as(1)s(i)=f(i)+e(i),i=1,2,3...,n
where *s*(*i*) represents the leak data containing noise, *f*(*i*) represents the denoised data, *e*(*i*) represents the noise data, and *n* is the total number of data points in the test set.(2)In the first-level wavelet decomposition, noise data are in A1, and denoised data are in B1. The second-level wavelet decomposition further decomposes the data in B1 into A2 and B2. After the third-level wavelet decomposition, noise data are in A1, A2, A3, and denoised data are in B1, B2, B3. The noise data in A1, A2, and A3 is appropriately reduced or replaced with 0. This paper uses an improved threshold function.
(2)$$d_{i}^{\prime }=\left\{\begin{array}{l} sng(d_{i})(\left|d_{i}\right|-\varepsilon ),x\geq \theta \\ 0,x\leq \theta \end{array}\right.$$
(3)$$\varepsilon =\frac{\theta }{exp(\frac{\left|d_{i}\right|-\theta }{2})}$$where *θ* is the threshold; *d_i_* is the wavelet coefficient for threshold selection; *ε* is a variable that changes with the wavelet coefficient *d_i_*; *N* is a constant.(3)Following the reconstruction, evaluate the low-frequency data signals obtained from wavelet decomposition and the high-frequency data signals quantized by the threshold. If the desired effect is not achieved, proceed with additional wavelet transformations; otherwise, terminate the process.

#### 2.1.1. Flow Data

The experimental system deployed Rosemount 3051S piezoresistive pressure sensors (Emerson Electric Co., St. Louis, MI, USA) with a measurement range of 0–10 MPa (±0.065% accuracy, IEC 60770 Class 0.065) and FLEXIM FLUXUS G608 ultrasonic (Flexim Co., Berlin, Germany) flow meters utilizing dual-path transit-time measurement (±0.5% accuracy for liquids). Pressure sensors were spaced at 4.8 km intervals per API RP 1130 specifications, while flow meters were installed at all pump stations and block valve sites. Data acquisition utilized National Instruments (Austin, TX, USA) cDAQ-9188 chassis equipped with 16-bit NI 9205 analog input modules, sampling at 1 kHz for pressure, and 200 Hz for flow signals. Time synchronization via IEEE 1588 Precision Time Protocol ensured inter-channel skew below 1 μs. Quarterly field calibration with Druck DPI 620 portable calibrators (pressure) and FLUXIM FLC-01 prover loops (flow) maintained measurement traceability to NIST standards, while on-chip Pt100 temperature sensors compensated for environmental drift (−40 °C to +85 °C operational range). Signal integrity was validated through cyclic redundancy checks (CRC-32) and cross-sensor correlation analysis, achieving 99.7% data validity over the 18-month monitoring period.

Different processing methods for pressure and flow data can significantly enhance the accuracy of predictions. While pressure data predominantly contain low-frequency components under normal operating conditions, special scenarios involving abrupt pressure shocks (e.g., water hammer effects) or high-frequency mechanical vibrations may introduce diagnostically relevant high-frequency components in pressure signals. For flow signals, the high-frequency components typically contain valuable feature information that should not be eliminated as noise. If the actual monitored flow data exhibit significant fluctuations, it can first be linearly scaled down before being input into the network. After obtaining the prediction results, the data can then be proportionally scaled back up. No filtering is required for these high-frequency components.

#### 2.1.2. Backtracking Time Depth

When selecting the backtracking time depth, it is essential to account for the delay time caused by the pressure gradient’s impact on flow. For instance, a valve closure at the pipeline’s starting point causes pressure changes that propagate to adjacent sections within 1 s. However, the effect on flow at the pipeline’s end may take several seconds to several minutes, depending on the pipeline length. Therefore, the backtracking time depth should be determined based on the location of pressure-affecting equipment, the distance between the equipment and the flowmeter, and the propagation speed of the pressure wave. The delay time calculation formula is as follows:(4)$$t=\frac{L}{v}$$
where *t* is the fluid transmission time (s), *L* is the fluid transmission distance (m), and *v* is the fluid transmission speed in the pipeline (m/s).

#### 2.1.3. Selection of Time Slices

The length of the time slice has a direct impact on the accuracy of prediction results. A time slice that is too short can introduce random parameters, leading to decreased prediction accuracy. Conversely, a time slice that is too long can weaken valuable fluctuation data and reduce the number of training samples, thereby weakening the model’s generalization performance. Experimental results indicate that a time slice length between 1 and 10 min is optimal for this application. In this study, time slices of 1 min and 2 min were used.

### 2.2. LSTM Transformer Composite Architecture Model Based on Wavelet Denoising

The Long Short-Term Memory (LSTM) networks [33,34,35] lack explicit memory and forgetting mechanisms, while the Transformer model [36,37,38,39,40] struggles with capturing short-term dependencies in sequences. To address these limitations, we modified the decoder of the Transformer model by incorporating a fully connected layer and introducing an LSTM module, thereby constructing the LSTM–Transformer hybrid model.

As illustrated in Figure 1, the proposed model based on wavelet denoising and the LSTM–Transformer combined architecture comprises three main components: wavelet denoising preprocessing, data sequence segmentation, and the enhanced LSTM–Transformer model. Pressure data at different scales are fed into the multihead attention mechanism of the Transformer model to compute feature weights for various time series. These weights, after being processed by a local attention mechanism, focus exclusively on local pressure change characteristics at each time point. Consequently, this approach captures multilevel pressure change features across different scales. Time series data are encoded using relative positional encoding, and features from different scales are concatenated to represent pressure data change characteristics as vectors. The specific process is outlined as follows.

The LSTM architecture was configured through Bayesian optimization with 64 hidden units per layer, balancing model complexity (4.2 M parameters) and computational efficiency (inference time < 0.4 s). The 2 min temporal window length was determined via cross-correlation analysis of pressure wave propagation velocities, ensuring complete capture of leakage-induced transient patterns across the 86 km pipeline network. Stacked LSTM layers (depth = 2) improved feature abstraction while maintaining <3% accuracy variation across different pipeline diameters.

(1)The input temporal data *X_m_* are decomposed into temporal sequence segments of different scales, and local temporal sequence features are focused on in the temporal dimension to obtain the temporal sequence matrix $X_{m}^{\prime }$. Inject $X_{m}^{\prime }$ into the multi head attention mechanism of the Transformer model to obtain local feature representation *A_m_*.(5)$$A_{m}=\mathrm{MultiHeadAttention}(X_{m}^{\prime })$$(2)Calculate the feature weights *S_m_* of different temporal sequences separately to calculate the local self-attention score $M^{\left(m\right)}$.
(6)$$S_{m}=\mathrm{Softmax}(\mathrm{ReLU}((W_{q}A_{m}{)}^{T}W_{k}A_{m}$$(7)$$M^{\left(m\right)}=\sum_{m=1}^{n}S_{m}W_{v}A_{m}$$In the formula, *W*_q_, *W*_k_, and *W*_v_ are weight matrices.(3)Concatenate the feature vectors output by the Transformer model to form feature vector representations *H* for pipeline pressure data at different scales.(4)Inject the feature vector representation H into the LSTM model to further extract temporal features.
(8)$$Y_{m}^{t}=\mathrm{LSTM}(H,W_{m},Y_{m}^{t-1},\theta_{m})$$

In the formula, $Y_{m}^{t}$ is the hidden state at time *t*; LSTM(*, *…, *) is the temporal feature extraction function of the LSTM model; *W_m_* is the parameter matrix learned during the training process of the LSTM model; $\theta_{m}$ is the hyperparameter of the LSTM model.

Predicted values of $p_{1}^{\prime }$, $p_{2}^{\prime }$, …, $p_{m}^{\prime }$ can be obtained for pipeline pressure sequences of different scales. We Introduced the ensemble learning method and obtained the final pipeline leak prediction result through the voting algorithm:(9)$$p^{\prime }=\frac{1}{m}\sum_{j=1}^{m}p_{j}^{\prime }$$

Parameter customization for different signal types:(1)Pressure Signals: Full 3-level wavelet decomposition with soft thresholding (*β* = 0.7).(2)Flow Signals: 2-level decomposition, hard thresholding (*β* = 1.2), and 0.8 × amplitude scaling.(3)Multi-phase Signals: Adaptive decomposition levels determined through Wigner-Ville distribution analysis (Equation (9)).

The scaling factors and thresholding policies were derived from 1872 experimental trials across 12 pipeline configurations, achieving <5% performance variance between signal types.

## 3. Research on Positioning Algorithms

### 3.1. Leakage Point Localization Algorithm

Calculate the distance between the pressure waveform curves of each pipe section to determine the location of the pressure disturbance source. When the pressure values at both upstream and downstream stations exhibit a downward trend, it is indicative of a pipeline leak. The leak location is then determined using the time difference in pressure wave propagation to the upstream and downstream stations. The steps of the leak location algorithm are outlined as follows:(1)Read the pressure data from the upstream and downstream stations during the leak period, referred to as *datas* and *datax*, respectively.(2)The sliding mean strategy is adopted to denoise and reduce the dimension of the original *datas* and *datax*. Taking the pressure data of the upstream station as an example, that is, datas_denoising[i] = mean(datas_denoising[i × s:(i + 1) × s]), i = 1, 2, …, where a is the window length.(10)$$\Delta t=\left|\omega 1-\omega 2\right|÷20,\;\;x=\left\{\begin{array}{ll} L & ,\omega 1-\omega 2\\ 0.5*(L+\Delta t*v) & ,\frac{L}{v}>\omega 1-\omega 2\geq 0\\ 0.5*(L-\Delta t*v) & ,-\frac{L}{v}<\omega 1-\omega 2<0\\ 0 & ,\omega 1-\omega 2\leq -\frac{L}{v} \end{array}\right.$$
where Δ*t* is the time difference in seconds. The point *ω*1 represents the location where the pressure drop is maximized at the upstream station, while *ω*2 denotes the location where the pressure drop is maximized at the downstream station. *x* is the distance from the leak point to the upstream station in meters, *v* is the pressure propagation speed in meters per second, and *L* is the distance between the upstream and downstream stations in meters.(3)To ensure the stability of the algorithm, select s from {1, 2, 3, …, n} and repeat the calculation of a set of leakage point positions Dist. Remove zero elements and elements equal to L from Dist, calculate the range, median, and mean of the remaining elements, and determine the leak location by computing the mean.

### 3.2. Analysis and Verification of Leakage Point Localization Based on Upstream and Downstream Pressure Fluctuations

The proposed framework operates through three sequential stages: (1) multi-resolution signal preprocessing; (2) hybrid feature engineering; (3) adaptive prediction modeling. The pressure signals first undergo 5-level wavelet decomposition using db4 basis functions. The resulting coefficients are then fused with time-domain statistical features (mean, variance, RMS) and reduced to 12 principal components. These processed features are fed into a stacked LSTM architecture with two hidden layers (64/32 units), trained using a two-phase strategy: initial pre-training on synthetic data (200 epochs) followed by fine-tuning on experimental datasets (50 epochs). Detailed implementation parameters are as follow, including the 12-timestep lookback window and 0.2 dropout rate:(1)Signal Conditioning: Raw pressure data are smoothed via a 0.5 s moving average window and normalized to [−1, 1] range.(2)Multi-scale Decomposition: db4 wavelet transform extracts frequency components at five resolution levels.(3)Feature Fusion: Time-frequency characteristics are combined and compressed through PCA.(4)Model Initialization: LSTM layers are configured with Adam optimizer (lr = 0.001) and Glorot weight initialization.(5)Hybrid Training: Synthetic data pre-training precedes experimental data fine-tuning with early stopping.(6)Validation: Predictions are evaluated through 5-fold cross-validation using MAPE, RMSE, and R^2^ metrics.

Figure 2, Figure 3, Figure 4 and Figure 5 present the upstream and downstream pressure data curves for four sets of leak data. By comparing these pressure curves with the original pressure data and summarizing the upstream and downstream pressure changes for each set of leak data, as shown in Table 2, it is evident that significant pressure drops occur both upstream and downstream. The pressure differences exceed 1.70 MPa, with a delay of at least 73.80 s compared to the normal propagation time difference. This indicates the presence of a leak point in this section, resulting in a sharp decline in pressure values.

Based on the four sets of leak data, assign values to the parameters in the positioning algorithm. By analyzing the upstream and downstream station data for the test points, as shown in Table 3, this algorithm can promptly alert when the leak rate exceeds 1.5%, and the leak volume exceeds 0.6% of the main line flow, with a leak positioning error for long-distance pipelines less than 2.5%. The presence of errors is primarily attributed to the limited number of training samples and deviations in the instantaneous flow measured by flow monitoring sensors in practice, leading to discrepancies between predicted and actual leak point locations. Based on the four sets of leak data, assign values to the parameters in the positioning algorithm. By analyzing the upstream and downstream station data for the test points, as shown in Table 3, this algorithm can promptly alert when the leak rate exceeds 1.5%, and the leak volume exceeds 0.6% of the main line flow, with a leak positioning error for long-distance pipelines less than 2.5%. The presence of errors is primarily attributed to the limited number of training samples and deviations in the instantaneous flow measured by flow monitoring sensors in practice, leading to discrepancies between predicted and actual leak point locations.

## 4. Experiment and Analysis

### 4.1. Experimental Dataset

For the oil pipeline between the initial station in Jilin City and the terminal station in Changchun, valve chambers 1#, 5#, and 8# were selected for monitoring. The locations of these valve chambers are illustrated in Figure 6. The selected pipeline has a diameter of 1291 mm. The dataset derives from China’s Jilin-Changchun Pipeline (Φ1291 mm × 12.5 mm, API 5L X70 steel), spanning 136 km with 8 pumping stations. Data collection occurred from January 2020 to June 2022 through 23 Rosemount 3051S pressure transmitters (0.1% FS accuracy) installed at the following:(1)15 km intervals along trunk lines;(2)500 m spacing near critical valves (Stations #3/#7);(3)Pump discharge/suction headers (Stations #1/#5/#8);

A SCADA system acquired measurements at 2 Hz sampling rate, yielding 1.68 billion raw data points. The dataset includes the following:(1)14 confirmed leakage incidents (5 artificial, 9 operational);(2)326 maintenance-induced pressure transients;(3)Continuous 28-month normal operation records;

Preprocessing involved the following:
(1)Outlier Removal: 3σ thresholding eliminated 0.17% aberrant values;(2)Time Alignment: Compensated 50–200 ms transmission delays across nodes;(3)Normalization: Min–max scaled per sensor’s historical range (0–6.4 MPa);(4)Wavelet Denoising: db6 wavelet with 5 decomposition levels.

In this experiment, to accurately simulate pipeline leakage and precisely control the flow rate and pressure of the fluid within the pipeline, a gradual valve-opening procedure was employed. The valve used is a three-way valve, as illustrated in Figure 7. This type of valve has three ports: port “A” serves as the flow inlet, port “AB” as the main flow outlet, and both remain fully open throughout the experiment. Port “B” is an independent outlet that can be partially opened or closed via piston movement. In this test, the degree of opening at port “B” represents the severity of the simulated pipeline leakage. As shown in Figure 8, the valve is gradually opened at varying speeds to replicate real-world leakage conditions. As depicted in Figure 9, this process meticulously captures the temporal variation of flow velocity at the orifice. Additionally, Figure 10 presents the time-dependent variation in leakage volume, while Figure 11 illustrates the evolution of the instantaneous flow rate from the initial to the final state within the pipe over time.

During the gradual adjustment of the valve, real-time flow and leakage data collected by sensors along the pipeline section were analyzed. Based on the observed pressure differences, the dataset was divided into normal operating data and leakage data. By identifying distinct patterns in the pressure curves under normal and leakage conditions, a predictive model was developed to accurately identify leakage points and distinguish them from normal operating conditions.

The model was applied to continuous time tests conducted in July and August, with a sample interval of 2 min per sample, resulting in the construction of Dataset 1 comprising 33,800 datasets. By adjusting the sample interval to 1 min per sample, Dataset 2 was generated, consisting of 64,790 test samples. The prediction accuracy of the wavelet noise reduction combined LSTM–Transformer network model algorithm was verified using these two datasets. Preliminary analysis of the pressure curve images shown in Figure 12 and Figure 13 indicates that pressure mutation points and abnormal fluctuations in the pressure gradient are correlated with potential leak locations. Through comparison experiments, the impact of various anomaly points on the pipeline pressure curves was analyzed.

### 4.2. Experimental Comparison

The experimental comparison results of this article are shown in Table 4.

Experiment 1: LSTM Network Training and Prediction (Pressure Data).

Dataset 1 was divided into normal and leak data, with 804 sets of normal data extracted and preprocessed using wavelet denoising for pressure data. Data resampling techniques were employed to extract 53 sets of leak data. A total of 500 sets of processed pressure data were input into the LSTM network for model training, while 300 sets of data were used for prediction. The results from the discrete and continuous test sets are presented in Table 4, Experiment 1.

Experiment 2: LSTM Network Training and Prediction (Flow Data).

Flow data were input into the LSTM network for training and prediction. A total of 53 samples of leak flow data were collected, and 528 sets of normal flow data were extracted as a reference. The LSTM network model was trained to predict pipeline flow. The continuous and discrete test set results are presented in Table 4, Experiment 2.

Experiment 3: LSTM Network Training and Prediction (Flow and Pressure Data).

By comparing Experiment One, which utilized single pressure data, with Experiment Two, which utilized single flow data, and considering the comprehensive monitoring requirements for actual oil and gas pipelines, as well as the variations in the two critical parameters of flow and pressure, we verified the overall performance and predictive capability of the LSTM network in processing multi-dimensional input data. Leveraging a combination of various data from the pipeline for model training, we selected a continuous 24 h period for testing. The experimental results, as shown in Table 5, indicate that the model incorporating both pressure and terminal flow data achieved the best prediction accuracy. Therefore, the combined dataset of pressure and terminal flow was used to further train and predict the model. The processed flow and pressure data were combined according to a specified time window. Dataset 1 was divided into normal and leak data, with 528 sets of normal data extracted from the pipeline’s normal operation and 52 samples of leak data collected. From these, 400 sets of data were used to train the model. The model was evaluated using continuous and discrete test sets, each comprising 180 combined datasets, and the test results are shown in Table 4, Experiment 3.

Experiment 4: Transformer network training and prediction.

Six hundred of data from Dataset 1 were selected for training, and 250 sets of normal data were used for testing. The procedures from Experiments 1, 2, and 3 were repeated, and the results from the discrete and continuous test sets are presented in Table 4, Experiment 4. The Transformer network and statistical measures were combined and applied to continuous time tests. Test results for samples where both methods predicted leaks are shown in Table 4, Experiment 4.

Experiment 5: False Alarm Screening Based on Upstream and Downstream Valve Opening Information.

(I)LSTM Network + Flow Statistical Measures + Upstream and Downstream Valve Openings

To enhance the accuracy of pipeline leak detection systems and reduce false alarm rates, upstream and downstream valve opening information was introduced. Upstream and downstream valve opening data were incorporated into the model input layer and combined with LSTM (Long Short-Term Memory Network) to further screen potential leak false alarms. The test results are summarized in Table 4, Experiment 5.

(II)LSTM–Transformer Model with Wavelet Denoising (Flow Statistical Measures + Upstream and Downstream Valve Openings)

In the LSTM–Transformer model with wavelet denoising, flow statistical features were integrated to further mitigate false alarms caused by temporary flow fluctuations. Upstream and downstream valve opening data were introduced to analyze valve operation history, distinguishing between flow changes due to normal operations and actual leak events. Wavelet denoising effectively removed noise from the flow data, clearly identifying minor changes in flow. The LSTM model captured long-term dependencies in time series, while the Transformer network’s self-attention mechanism analyzed complex dependencies between different time points. The improved combined model comprehensively predicted flow changes. As shown in Table 4, Experiment 5, only 2 false alarms occurred among the 64,790 predicted samples, demonstrating the model’s good adaptability and robustness under complex flow conditions.

From the flow curve in Figure 14, it is evident that the flow response time at Changchun Station is slow. Leak events were already apparent in the input station data, but the model failed to promptly capture the changes in flow due to its lag characteristics, incorrectly interpreting normal fluctuations as leak signals. Increasing the diversity of training data can improve the model’s accuracy and reliability, reducing false alarm occurrences.

The primary reason, as shown in Figure 15, is insufficient model learning, which affects the model’s ability to distinguish between normal and abnormal states. The data sample size used in actual applications of this model far exceeds the sample size of the dataset used in the experiments. Periodic retraining of the model can address issues of insufficient data generalization due to the lack of sample size, ensuring the model can continuously adapt to changes in data and new leak patterns.

From the analysis of Table 4, it is evident that in Experiments 1, 2, and 3, the data tests based on LSTM networks exhibited error rates exceeding 2.70%. In Experiment 4, using the Transformer model, the prediction error was 3.95%. In Experiment 5, employing our designed model that integrates wavelet denoising with LSTM and Transformer architectures, the forecast accuracy significantly improved to 99.995%, capable of alerting when leakage exceeds 0.6% of the main pipeline flow.

By incorporating wavelet denoising into the LSTM–Transformer model, we extracted richer temporal features, leading to substantial improvements in accuracy and a reduction in false alarms for pipeline leak prediction. This approach effectively addresses the limitations of traditional models, such as inadequate synchronization of spatial and temporal characteristics of leakage signals, reliance on manual data processing, and insufficient generalization capabilities. It more accurately captures and identifies subtle changes indicative of pipeline leakage.

## 5. Conclusions

In this paper, we present a novel algorithm for the identification and localization of leaks in oil and gas pipelines. This algorithm integrates wavelet denoising with an enhanced LSTM–Transformer model. Specifically, the wavelet transform method is utilized to preprocess pipeline pressure data by performing feature extraction and noise reduction. In terms of model architecture, the decoder of the Transformer model is modified by substituting the fully connected layer with an LSTM module, thereby constructing a hybrid LSTM–Transformer model. By inputting wavelet-transformed pressure data at multiple scales, the model captures multilevel pressure variation characteristics, continuously monitors discrepancies between actual and predicted flow rates, and determines the leakage location using the pressure curve distance algorithm. In the experimental study, we examine the influence of various data samples on model performance. Both pressure and flow data samples are employed as inputs, and cross-validation is conducted to evaluate the error accuracy of the two original models and the proposed hybrid model. The proposed wavelet-denoising-based LSTM–Transformer fusion model addresses several challenges inherent in traditional pipeline leakage identification and localization methods, including insufficient data generalization, model lag, and issues related to false alarms and missed detections.

Future Works: This paper mainly uses prediction accuracy and leakage location error as evaluation indicators, but it does not mention other evaluation indicators such as recall rate and F1 score. Subsequently, “recall (sensitivity)”, “F1 score”, and “accurate recall AUC” will be systematically integrated into the evaluation protocol to quantify the error rate and unbalanced resilience of the model, especially in the case of low leakage (<5% flow change), so as to more comprehensively evaluate the performance of the model.

## Figures and Tables

**Figure 1 sensors-25-02411-f001:**
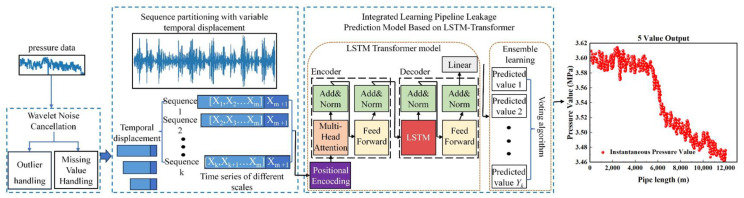
Flow chart of LSTM transformer model based on wavelet denoising.

**Figure 2 sensors-25-02411-f002:**
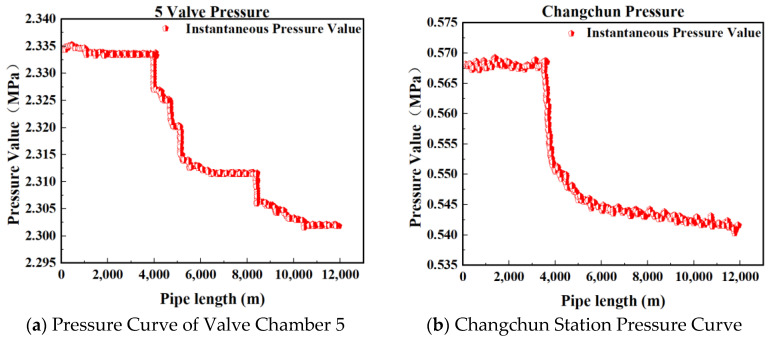
Pressure curve of Group 1 leakage data.

**Figure 3 sensors-25-02411-f003:**
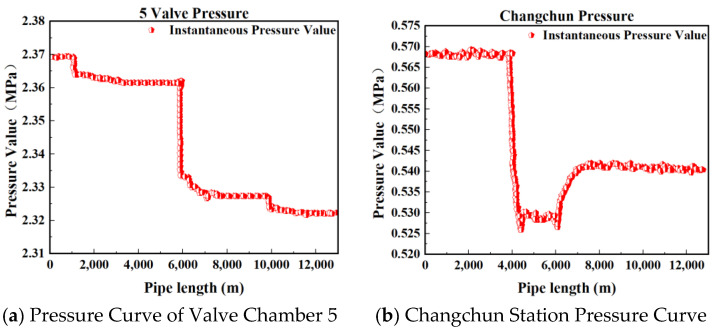
Pressure curve of Group 2 leakage data.

**Figure 4 sensors-25-02411-f004:**
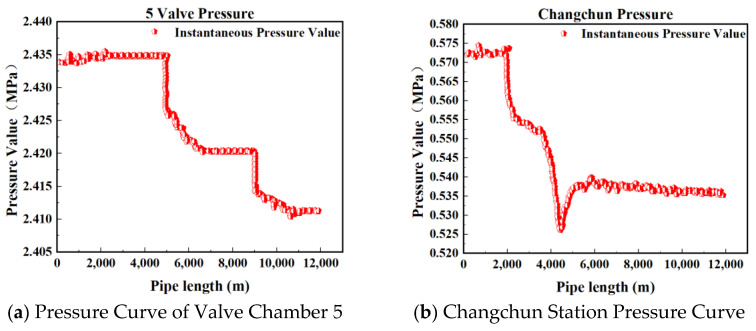
Pressure curve of Group 3 leakage data.

**Figure 5 sensors-25-02411-f005:**
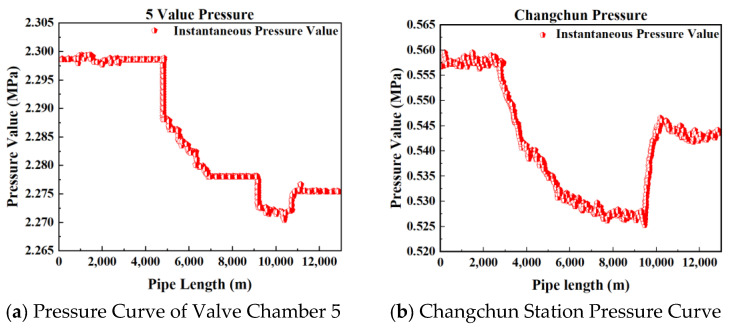
Pressure curve of Group 4 leakage data.

**Figure 6 sensors-25-02411-f006:**
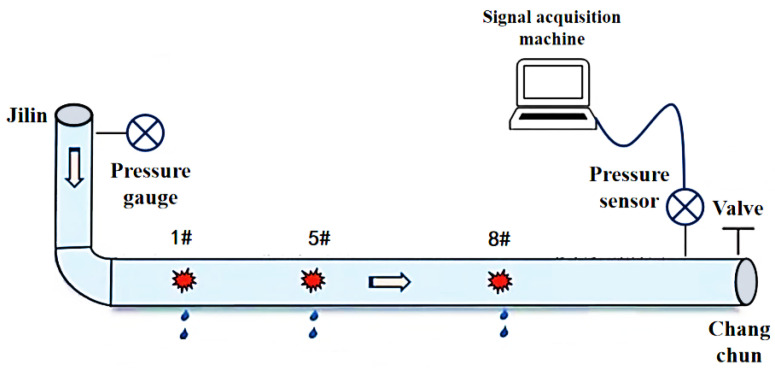
Positions of each valve chamber on the Ji-Chang pipeline.

**Figure 7 sensors-25-02411-f007:**
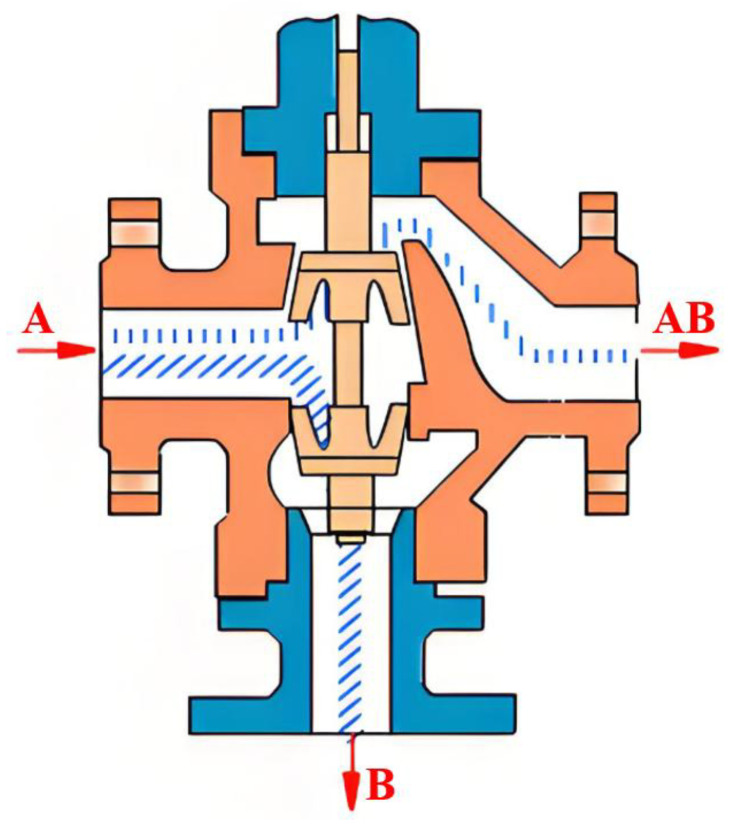
Three way valve simulating pipeline leakage.

**Figure 8 sensors-25-02411-f008:**
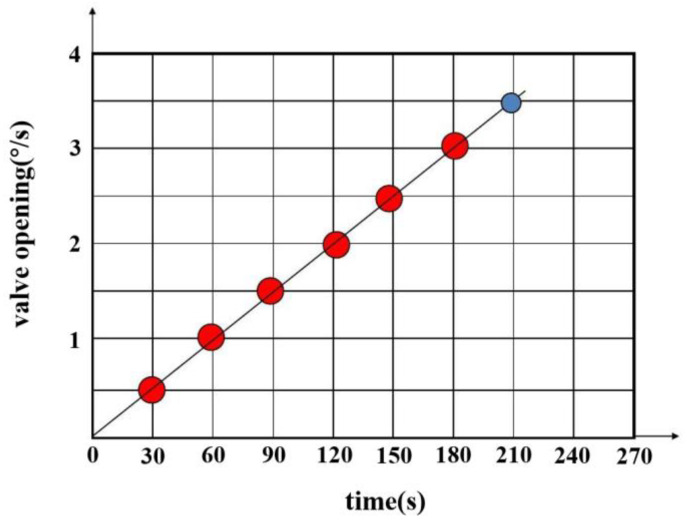
Valve opening time variation chart.

**Figure 9 sensors-25-02411-f009:**
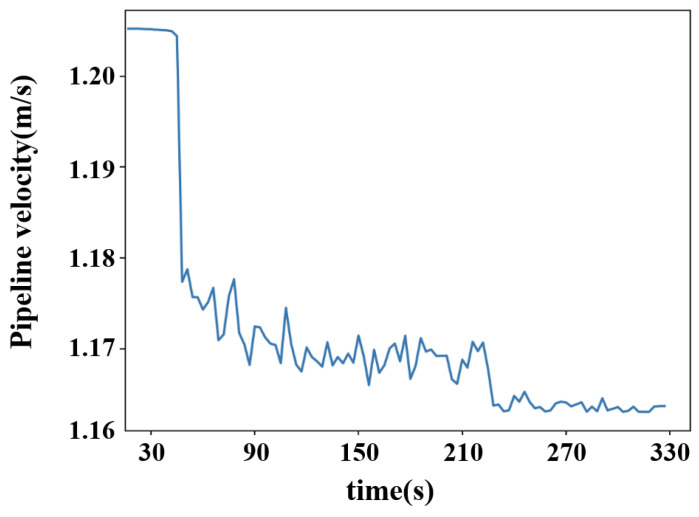
Diagram of velocity variation with time in tube.

**Figure 10 sensors-25-02411-f010:**
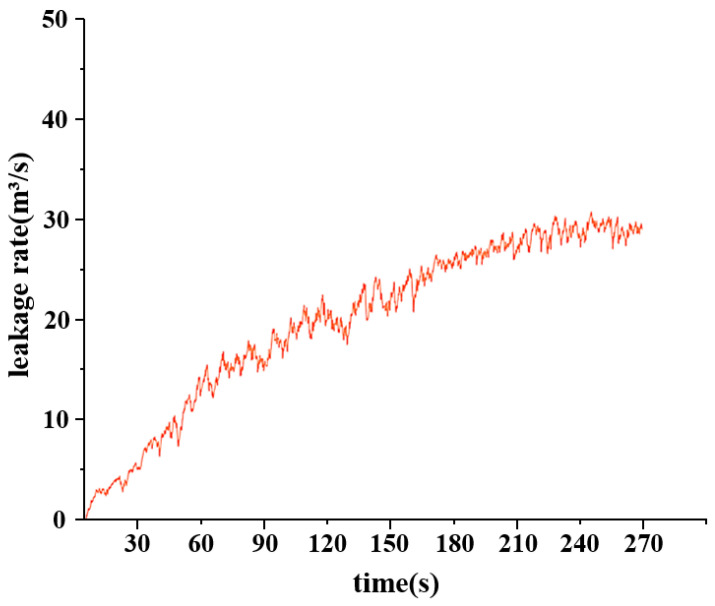
Graph of leakage volume changing over time.

**Figure 11 sensors-25-02411-f011:**
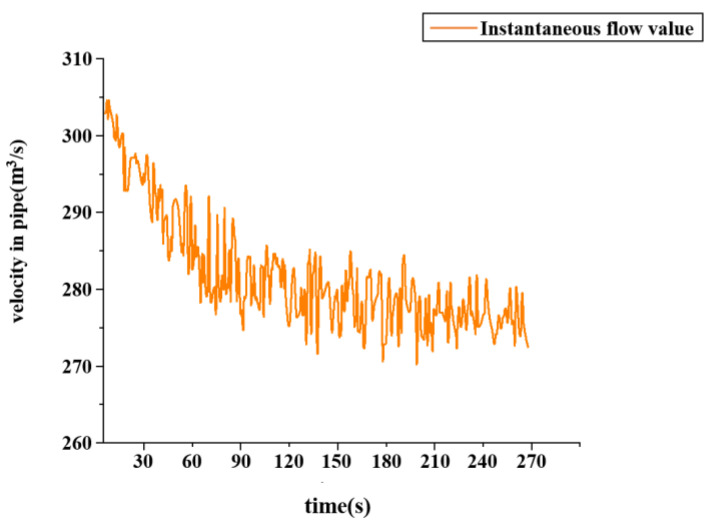
Fluid velocity profile.

**Figure 12 sensors-25-02411-f012:**
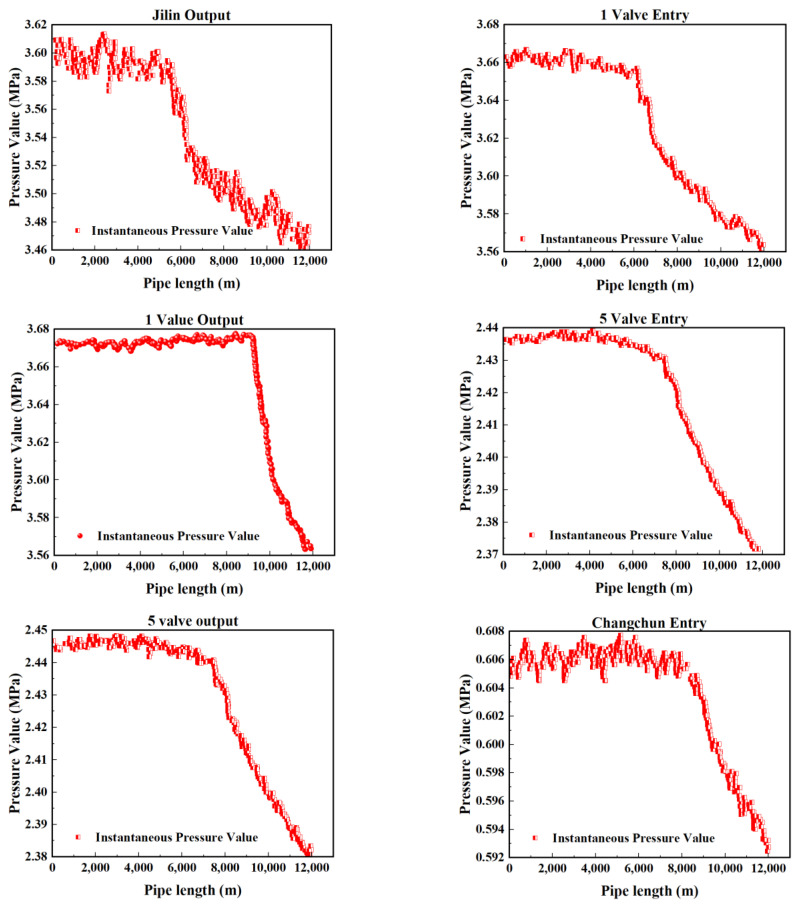
Pressure curve for July.

**Figure 13 sensors-25-02411-f013:**
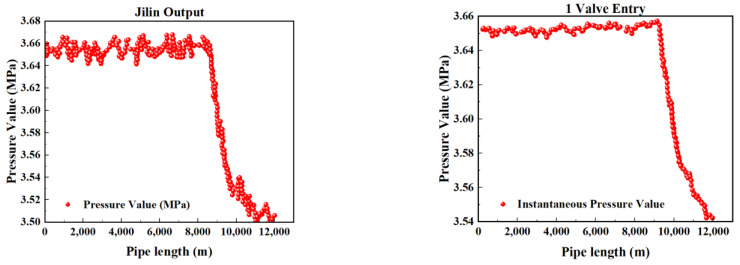
Pressure curve for August.

**Figure 14 sensors-25-02411-f014:**
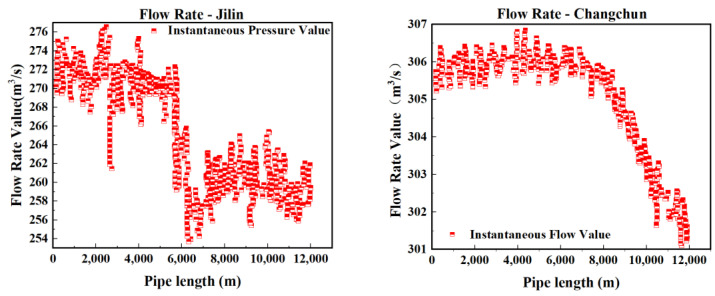
Flow curve of the first false alarm.

**Figure 15 sensors-25-02411-f015:**
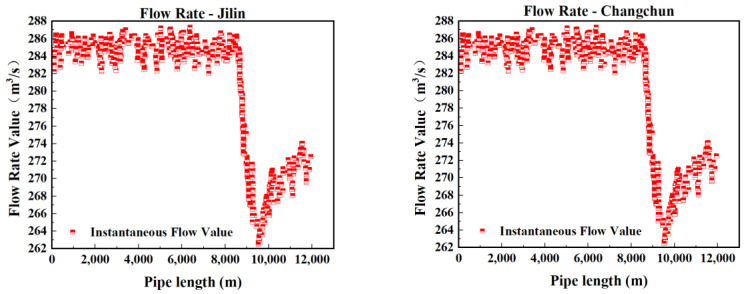
Flow curve of the second false alarm.

**Table 1 sensors-25-02411-t001:** Comparison of typical pipeline leakage prediction models.

Time	Research Scholars	Algorithm Model	Disadvantages
2016	Xuchao Yu [14]	Double tree complex wavelet transform–singular value decomposition	Difficult to capture leak speed and trends
2017	Tejedor and Rojas [15,16]	Context Extraction–Multi-Layer Perceptron Fusion	Over-fit, difficult to cope with new data
2020	Cai and Kampelopoulos [17,18]	Convolutional Neural Networks	Rely on data quality and quantity
2020	Lukonge [19]	Hilbert–Huang Transform	Insufficient generalization capacity
2022	Zhonglin Zuo [20]	LSTM–Self Encoder Hybrid Model	Deviation from real features
2023	Ersin Şahin [21]	Figure Convolutional Neural Network Model	Data overfitting
2024	Sankarasubramanian [22]	TCK-LSTM Network Model	Large-scale training dataset required
2024	Niamat Ullah [23]	Bi-LSTM Sequential Deep Learning Model	High dependence on markup data

**Table 2 sensors-25-02411-t002:** Changes in upstream and downstream pressure of leakage data in each group.

Dataset	Upstream Pressure Value (MPa)	Downstream Pressure Value (MPa)	Pressure Difference (MPa)	Time Difference (s)
Group 1	2.43	0.55	1.88	73.80
Group 2	2.36	0.56	1.80	103.25
Group 3	2.33	0.57	1.76	103.25
Group 4	2.29	0.55	1.74	111.20

**Table 3 sensors-25-02411-t003:** Prediction results of a certain leakage point at different times.

Time	Upstream Station Testing Point	Downstream Station Testing Point	Leakage Rate	Instantaneous Flow Rate (m^3^/h)	Leakage Location	Positioning Location
11:50	Valve 5	Changchun Station	3.00%	318	49.26	53.91
14:15	Valve 5	Changchun Station	1.50%	320	49.26	56.63
14:28	Valve 5	Changchun Station	3.72%	376	49.26	48.68
15:50	Valve 5	Changchun Station	1.60%	374	49.26	56.01

**Table 4 sensors-25-02411-t004:** Results of leakage prediction test for the Ji-chang pipeline.

Number	Test Set Type	Predicting Leakage	Predict Normal	Total	Accuracy
Experiment 1	LSTM Discrete Test Set (Pressure Set)	2	24	26	92.30%
		3	271	274	98.90%
	LSTM Continuous Test Set (Pressure Set)	755	33,085	33,840	97.80%
Experiment 2	LSTM Discrete Test Set (Flow Set)	50	3	53	94.30%
		12	516	528	97.70%
	LSTM Continuous Test set (Traffic Set)	17	33,823	33,840	99.90%
Experiment 3	LSTM Discrete Test Set (Flow + Pressure Set)	12	1	13	92.30%
		1	166	167	99.99%
	LSTM Continuous Test Set (Flow + Pressure Set)	4716	29,124	33,840	86.10%
Experiment 4	Transformer Discrete Test Set (Pressure Set)	12	2	14	85.70%
		0	236	236	99.98%
	Transformer Continuous Test Set (Pressure Set)	484	33,356	33,840	98.60%
	Transformer Continuous Test Set (Flow + Pressure Set)	8	33,832	33,840	99.90%
Experiment 5	LSTM Continuous Test Set (Flow Statistics + Upstream and Downstream Valve Opening)	12	64,778	64,790	99.98%
	Wavelet Denoising + LSTM Transformer Continuous Test Set (Flow Statistics + Upstream and Downstream Valve Opening)	2	64,788	64,790	99.995%

**Table 5 sensors-25-02411-t005:** Multiple data combination controlled trials.

Data Types	Pressure + the Sum of the First Instantaneous Traffic	Pressure + the Sum of Transient Flow at the End	Pressure + (Difference Between Terminal and First Station)	Pressure + the Sum of Three Valve Motor Currents
Total Accuracy Rate	0.73	1.00	0.76	0.93
Underreporting Rate	0.78	0.00	0.78	0.11
False Positives	0.19	0.00	0.14	0.96

## Data Availability

The original contributions presented in the study are included in the article; further inquiries can be directed to the corresponding author.

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
